

# Collapse of Deep-Sea Circulation during an Eocene Hyperthermal Hothouse – A DeepMIP Study with CESM1.2

Arne M.E. Winguth[1], Mikaela Brown[1], Pincelli Hull[2], Elizabeth Griffith[3], Christine Shields[4], Ellen Thomas[2], and Cornelia Winguth[1]

[1]Department of Earth and Environmental Sciences, University of Texas Arlington, 500 Yates St., Arlington, TX 76016, USA.
[2]Department of Geology and Geophysics, Yale University, P.O. Box 208109, New Haven, CT 06520-8109, USA.
[3]School of Earth Sciences, The Ohio State University, South Oval Mall, Columbus, OH 43210, USA.
[4]National Center for Atmospheric Research, P.O. Box 3000, Boulder, CO 80307-3000, USA.

*Correspondence to*: Arne M.E. Winguth (awinguth@uta.edu)

**Abstract.** During the Paleocene-Eocene Thermal Maximum (PETM, ~56 Ma), a rapid injection of greenhouse gases (with isotopically depleted carbon) into the atmosphere led to a ~5 °C global temperature rise, ocean acidification, and perturbation of marine and terrestrial ecosystems. In this study, we carried out a series of DeepMIP climate sensitivity experiments t using the Community Earth System Model CESM1.2 to evaluate how changes in the radiative forcing could have contributed to Eocene hyperthermal events. An atmospheric change from $3xCO_2$ relative to pre-industrial levels (PAL) equivalent during the latest Paleocene to $6xCO_2$ PAL in response to a carbon input pulse of 1680 PgC resulted in equatorial warming to 36.9 °C consistent with proxy estimates. The lower equator-to-pole temperature gradient in this $6xCO_2$ PAL scenario as compared to the pre-industrial experiment with $1x CO_2$ PAL is due to the lack of an ice sheet, the increase in greenhouse gases, and a lower cloud optical depth. The climate simulations suggest an intensified hydrological cycle with higher precipitation in the tropics, particularly over the Indian Eocene continent, and in mid-latitude. In contrast, mega-droughts are prominent in the subtropics, particularly in Africa and South America. Topographic effects such as the closure of the Drake Passage and the more southern location of Australia as well as a lower-than-present meridional temperature gradient contribute to a much weaker surface ocean circulation near the Antarctic continent, as compared to the current pronounced Antarctic Circumpolar Current. In response to the increase in greenhouse gas forcing to $6xCO_2$ PAL, deep water formation in the Southern Ocean nearly collapsed and changed from a southern-dominated deep-sea ventilation to a weak deep water formation in the North Atlantic Ocean and further to a polar collapse of deep water formation and a shallow haline-mode ventilation in the subtropics at $12xCO_2$ PAL. Bipolar convective overturning in the Pacific Ocean is not supported and remains uncertain, but southern component water mass formation in the Pacific Ocean has been simulated with $1x CO_2$ PAL. Increased stratification and reduced solubility of dissolved oxygen caused by warming may have contributed to lower abyssal dissolved oxygen concentrations and thus stresses on the marine ecosystem. However, decreased upwelling and productivity may have decreased the apparent oxygen utilization and thus could have increased the oxygen concentration in the twilight zone.



## 1 Introduction

The generally warm climate of the early Paleogene (~65-45 Ma) was characterized by multiple, rapid, short-term global
warming events, termed "hyperthermals". Paleogene hyperthermal events were caused by massive greenhouse gas release into
the ocean-atmosphere, and impacted the Earth's ecosystems, biotic evolution, and global carbon cycle. As such, hyperthermals,
the most extreme of which was the 5°C - 8°C warming (Dunkley Jones et al., 2013; Lunt et al., 2017) during the Paleocene-
Eocene Thermal Maximum (PETM; ~56 Ma; Kennett and Stott, 1991; Westerhold et al., 2011), may be important for
understanding ongoing anthropogenic-induced global warming and its long-term effects on the environment.

The PETM is recognized as a major global carbon cycle disturbance (DeConto et al., 2012; Dickens, 2011; Thomas and
Shackleton, 1996; Röhl et al., 2007, Sluijs et al., 2007, Bowen et al., 2015). Estimates of the total amount of carbon released
during the PETM range from 3,000 to 10,000 petagrams (Pg) (e.g., Zachos et al., 2005; Zachos et al., 2008; Bowen et al.,
2015), potentially initiated by volcanism from the North Atlantic Igneous Province (Gutjahr et al., 2017), that could have
triggered disturbances of surface sedimentary carbon reservoirs, climate-carbon cycle feedbacks, associated widespread ocean
acidification (Zachos et al., 2005; Penman et al., 2014), and extinction of deep-sea benthic foraminifera (Thomas, 2007).

Past paleoclimate sensitivity studies (Lunt et al., 2012; Lunt et al. 2021; Inglis et al., 2020) generally reproduce PETM tropical
warming, a lower pole-to-equator temperature gradient, and reduced poleward heat-transport due to a rise in atmospheric $pCO_2$
(Winguth et al., 2010). Changes in poleward heat transport have also been attributed to the open central American seaway and
the closed Drake passage (e.g. Kenneth, 1973; Mikolajewicz et al. 1996, Sijp et al., 2009, Toumoulin et al., 2020), and open
passageways between the Atlantic and Artic Ocean (Shellito et al., 2009; Winguth et al., 2010). positive feedbacks on high-
latitude winter marine climate change involving convective clouds (Abbot et al., 2009), reduced cloud condensation nuclei
(CCN) (Kump and Pollard, 1998; Kiehl and Shields, 2013), or a generally warmer climate including hotter tropics (Huber,
2008).

In this study, we assess whether increases in atmospheric radiative forcing and associated changes in surface buoyancy forcing
are consistent with paleo proxies. Specifically, we focus on the contribution of changes in these forcings to a change in ocean
circulation at the onset of the PETM and the comparison of these simulated changes with circulation proxies.

## 2 Model Description

### 2.1 Community Earth System Model

In this study, the Community Earth System Model version 1.2 (CESM1.2) (Gent et al., 2011; Hurrel et al., 2013) was applied
with a horizontal resolution in the atmosphere and land of finite volume ~1.9°x2.5° with 26 layers, and in the ocean a nominal
resolution of 1°x1° with 60 vertical layers. Eocene boundary conditions follow the DeepMIP protocol (Lunt et al., 2017) as
summarized together with the model description. The model has been integrated for nearly ~ 2000 years to allow adjustment
of tracers in the deep sea.





## 1.2 DeepMIP Boundary and Initial Conditions

Boundary conditions for the PETM have been defined in DeepMIP (Lunt et al., 2017; Table 1). Paleo-topographic boundary conditions were modified from Herold et al. (2014) by including a subgrid topographic parameterization, tidal dissipation, river runoff, aerosols, and a land surface distribution to allow throughflow through narrow passages. Solar insolation ($S_0$ =1361 W m$^{-2}$) and orbital forcing (eccentricity $e$ = 0.06 and obliquity $\varepsilon$ = 23.5°) are kept at pre-industrial levels for all PETM experiments according to the DeepMIP protocol to allow a suitable comparison of the 1x$CO_2$ simulation with the preindustrial

experiment. It is assumed that the radiative forcing increase in solar radiation from the early Eocene to pre-industrial levels was compensated by the decrease in radiative forcing by the atmospheric methane concentration at that time.

Four sensitivity experiments with different $CO_2$ radiative forcings at 1x, 3x, 6x, and 12x preindustrial atmospheric levels (PAL) of 280 ppmv have been conducted according to the DeepMIP protocol (Table 1). In addition, a 1x$CO_2$ scenario with pre-industrial (1850) boundary conditions was carried out and compared to the 1x$CO_2$ Paleocene Baseline experiment with Eocene

topographic and ice sheet boundary conditions. The 3x$CO_2$ scenario represents conditions of the latest Paleocene (LP) before the onset of the PETM (or hyperthermal events), whereas the 6x$CO_2$ and 12x$CO_2$ PAL are carbon pulse sensitivity scenarios as inferred from the carbon isotope excursion (CIE).

Aerosol forcings likely were different than during pre-industrial conditions, thus could play an important role in climate change (see Kump and Pollard, 2008; Kiehl and Shields, 2013). We are using aerosol forcing from Herold et al. (2014) that utilizes

output from a Bulk Aerosol Model (Neale et al., 2010) and explicitly simulates the distribution of dust, sea salt, sulphate, and organic and black carbon aerosols consistent to the Eocene topography. The results from the 1x$CO_2$ PAL experiment with Eocene aerosol forcing are comparable to an additional sensitivity experiment at 1x$CO_2$ PAL with pre-industrial aerosol forcing (and thus not listed in Table 1).

The model has been initialized from a depth(z) and latitude($\varphi$)−dependent temperature profile (with a vertically declining

temperature T=25°C cos($\varphi$) (6000 z)/6000 +15°C for temperature z ≤ 5000 m and 15 °C for z > 5000 m) and from a vertically uniform salinity (S= 34.7) profile. All experiments have been integrated for 2000 years to near-equilibrium state and the top of the atmosphere energy balance has an absolute value below 0.55 W m$^{-2}$ for 1x$CO_2$ PAL and below 0.29 W m$^{-2}$ for 3x$CO_2$ PAL and higher $CO_2$ values which is slightly above the 0.147 W m$^2$ of CCSM4 at FV 0.9x1.25 atmospheric resolution.

## 3 Results

### 3.1 Simulated Paleocene Climate at 1x$CO_2$

Below, the Paleocene baseline scenario at pre-industrial $CO_2$ radiative forcing was compared to the preindustrial (PI) simulation with the same $CO_2$ radiative forcing (as part of the DeepMIP protocol) in order to explore the effects of topography, ice sheets, land surface, and aerosol forcing. A detailed summary of this Paleocene baseline simulation is given in Brown (2019). Differences in the global average surface air temperature (SAT) between the scenarios with Paleocene aerosol forcing



(1xCO$_2$ Paleocene Baseline, PB, scenario) and the scenario with 1xCO$_2$ Paleocene Baseline with preindustrial aerosol forcing (PB_PR scenario) are negligible.

The global average surface air temperature (SAT) is 7.4 °C higher for the 1xCO$_2$ Paleocene Baseline scenario than for the preindustrial simulations (Table 1; Fig. 1 and Fig. 2). SAT anomalies are amplified in polar regions (~44 °C over Antarctica, ~13 °C over the Arctic Ocean) and are linked to a lack of ice sheets, and surface and cloud albedo feedbacks. The difference

in surface elevation over Antarctica between the 1xCO$_2$ PETM scenario without ice sheets, and the preindustrial scenario exceeds 3km. Such an elevation change may result in ~18°C surface temperature change if a polar lapse rate of ~6 °C/km (as inferred from the 1xCO$_2$ Paleocene Baseline scenario) is assumed. The higher-than-pre-industrial surface air temperature in the 1xCO$_2$ Paleocene Baseline experiments leads to a reduced snow height and snow-free areas over West Antarctica, which in turn reduce the reflectivity and amplify the warming. Sea surface air temperatures in the tropics (20°S to 20°N) are ~30.6°C

in the 1xCO$_2$ Paleocene Baseline, 3.7°C warmer than in the preindustrial experiment, linked to a rise in surface temperatures and water vapor concentration in the atmosphere.

Global pole-to-equator precipitation patterns for the 1xCO$_2$ Paleocene Baseline scenario (Fig. 3 and Fig. 4) are comparable with the present-day, with precipitation higher than evaporation in the tropics and high-latitudes, and evaporation exceeding precipitation in the downward branch of the Hadley circulation cells. Global precipitation is ~18% higher in the 1xCO$_2$

Paleocene Baseline Scenario than in the PI experiment because of an increase in the latent heat flux from the surface to the atmosphere due to warmer polar temperatures, absence of ice sheets, and changes in the land surface albedo and thus enhanced water vapor in the atmosphere. The precipitation increases over land (by ~36%) in the 1xCO$_2$ experiment relative to the preindustrial conditions is linked to heavy rainfall over the tropics, due to the intensification of moisture transport along the lower branch of the Hadley Cell into the intertropical convergence zone (ITCZ; Fig. 5 and 6; Held and Soden, 2006). In

addition, enhanced precipitation is simulated over India (island effect), over northern South America with the opening of the Central American Seaway, and over West Africa due to the presence of an inland sea. Increased precipitation occurs at latitudes greater than 60° supporting a climate feedback associated with the hydrological cycle that may amplify the exceptional polar warmth (Pagani et al., 2006). Land-sea breezes may have influenced coastal wind patterns around Antarctica in this 1xCO$_2$ Paleocene Baseline scenario. The reduced air-sea temperature difference due to a lack of ice sheets contributed to a reduction

of the meridional temperature gradient and near sea ice-free conditions in proximity of Eocene Antarctica.

As a result of the lack of ice sheets and topographic changes, ocean currents around Antarctica in the 1x CO$_2$ Paleocene Baseline scenario were more sluggish than in the PI simulation. Closed or narrow gateways at the Tasman Sea and Drake Passage and an open Central American Seaway during the early Eocene, together with the reduced wind stress (Fig. 5) and weak meridional temperature gradient contributed to a substantially weaker-than-present ocean circulation, with the maximum

transport exceeding 180 Sverdrup (1 Sv = 10$^6$ m$^3$ s$^{-1}$ ) for the PI experiment as compared to 50 Sv for the 1xCO$_2$ PETM experiment (Fig. 5).

Compared to pre-industrial conditions, the low meridional temperature gradient and associated reduced wind stress as well as the open Central American Seaway led to weakening of the barotropic stream function in the North Atlantic Current by about





30 Sv (Fig. 5). In the southern hemisphere, the lack of an ice sheet, and thus a substantially weaker-than-preindustrial

meridional temperature gradient and associated reduced westerly wind stress, support a relatively slow westward-directed flow of 10 Sv-30 Sv around Antarctica, as compared to the pre-industrial strength of the westerly subpolar current around continental Antarctica, with values reaching nearly 200 Sv. In addition, the closed Drake Passage and Tasman Gateway and/or open Central America Seaway resulted in a relatively sluggish current around Antarctica during the early Eocene (Figure 5a). Reduced equatorward Ekman transport with declined wind stress produced favorable conditions for a deeper-than-preindustrial

mixed-layer depth in the proximity of the paleo-Weddell and Ross Sea, a higher-than-pre-industrial stratification in the North Atlantic Ocean (Fig. 6), and a global meridional overturning circulation with a dominant deep southern component and a shallow northern component (Fig. 7), consistent with previous studies (see for example Elsworth et al., 2017; Goldner et al., 2014; Inglis et al., 2015; Lefebvre et al., 2012; Mikolajewicz et al., 1993; Nong et al., 2000; Sijp et al., 2009; Zhang et al., 2010; Toumoulin et al., 2020).

There is almost no seasonal sea ice present around Antarctica in the $1xCO_2$ Paleocene Baseline scenario, whereas winter sea ice is simulated in the Arctic Ocean. Low salinity due to the wetter-than-present Arctic conditions and enhanced freshwater input from the adjacent land masses promotes sea ice formation during the boreal winter.

### 3.2 The $3xCO_2$ LP Scenario

The increase in $CO_2$ radiative forcing from $1xCO_2$ to $3xCO_2$ PAL leads to a substantial rise in the SAT and sea surface

temperature (SST) of 3.8 °C and 3.8 °C, respectively (Table 1). The 3x $CO_2$ latest Paleocene (LP) experiment simulates a 2.9 °C higher SAT in the tropics (20°N to 20°S) relative to the 1x $CO_2$ Paleocene baseline scenario, in response to the increase in $CO_2$ concentration and the associated climate feedbacks (Figs. 1 and 2a). The polar amplification of SAT (with an increase ranging from 8 °C to 13 °C) in the 3x $CO_2$ latest Paleocene scenario relative to the 1x $CO_2$ Paleocene reference case is linked to the ice albedo climate feedback. The north polar annual mean SAT in the 3x $CO_2$ latest Paleocene experiment is above

freezing (3°C), whereas SAT near the South Pole is below freezing (-5 °C) due to Antarctic continentality.

Differences in precipitation patterns relative to the $1xCO_2$ Paleocene baseline scenario follow future CESM1 climate change scenarios e.g. Meehl et al., 2013 with an increase in precipitation over the intertropical convergence zone, specifically over orographic features such as the Indian microcontinent, Northern Columbia, Western Central Africa, and Southeast Asia, and in subpolar latitudes along the westerly rain belts. High-latitude precipitation increased along mountain ranges on the windward

side of the westerly wind zones such as the Canadian Rockies and Patagonian Andes, and the West Antarctic Peninsula. Humid regions are associated with the ITCZ and midlatitude regions whereas arid regions are in proximity of the downward branch of the Hadley circulation. The hydrological cycle is amplified in the in $3xCO_2$ latest Paleocene experiment compared to the $1xCO_2$ Paleocene reference scenario, thus resulting in increased humidity in the tropics, particularly over India, and increased aridity over the subtropics due to an increase in evapotranspiration primarily in the Eastern Tethys region and southwest Africa.

Wind stress and barotropic stream function decreased relative to the $1xCO_2$ scenario, linked to polar amplification and a reduced meridional temperature gradient in response to an increase in sea-ice albedo, land snow cover, and associated



feedbacks. Trade winds (Fig. 5) and equatorial Ekman-induced upwelling declined by ~50% (Fig. 8), thus affecting surface temperature (Fig. 2a), net primary, and export production (Winguth et al., 2013).

Enhanced stratification with an increase in the $CO_2$ radiative forcing in the 3xCO2 LP scenario as compared to the 1xCO2

scenario led to a shallower mixed-layer depth layer (Fig. 6) and a weaker global meridional overturning cell, in particular a reduction in the southern component water masses (Fig. 7).

### 3.3 The 6xCO2 and 12xCO2PETM Scenarios

The sea surface temperature of the 6xCO2 PETM and 12xCO2 PETM climate simulations is 2.4 °C and 7°C respectively warmer than that of the 3xCO2 simulation (Table 1). Change in global SAT over land is comparably higher with a 4.3 °C and

10.4 °C respective rise due to multiple factors including polar decline in snow depth liquid water content equivalent, decrease in soil moisture, and associated lower heat capacity of soils and decline in latent heat flux. Changes in the Northern Hemisphere are even higher due to the land distribution, with a temperature increase above 7 °C.

Precipitation patterns in the 6xCO2 experiment are generally amplified (and even more so in the 12xCO2 experiment) compared to the 3xCO2 scenario in tropical and mid-to-high latitudes, because vapor moisture saturation increases ~7% per 1 °C increase

based on the Clausius-Clapeyron equation. Relative to the 3xCO2 scenario, there is a substantial increase in precipitation along the ITCZ, specifically over India and Northern South America near the Central American Seaway, linked to enhanced moisture transport. Compared to present-day, the zonal mean precipitation (Fig. 4) in the tropics increased by up to 913 mm/yr (2.5 mm/day) for the 6xCO2 experiment and by 1204mm/yr (3.3 mm/day) for the 12xCO2 experiment. Intensification of the down-ward branch of the Hadley cell and rise in temperature due to the increase in $CO_2$ radiative forcing relative to the 3xCO2

simulation enhance latent and sensible heat fluxes, thus leading to higher aridity and more severe drought conditions (Fig. 9). Wind stress and barotropic stream function are decreased relative to the 3x CO2 scenario, linked to polar amplification, and a reduced meridional temperature gradient in response to an increase in sea-ice albedo, land snow cover, and associated climate feedbacks. Trade winds (Fig. 5) and equatorial Ekman-induced upwelling declined by ~10-20% in the 6 x CO2 scenario and by 30-50% in the 12x CO2 scenario (Fig. 8), thus supporting an increase in equatorial surface temperature (Fig. 2 b) and a

decline in the net primary and export production with an increase in radiative forcing (Winguth et., 2013). Compared to the 3xCO2 latest Paleocene scenario, the mixed layer depth in the 6xCO2 and 12xCO2scenarios is further reduced (Fig. 6), leading to increased stratification and a collapse of the southern component deep water formation and meridional overturning circulation for the 12xCO2 scenario (Fig. 8), with a shift form the thermal polar mode of deep-water formation to a subtropical haline shallow mode.

In contrast to the 3xCO2 latest Paleocene scenario, the Arctic Ocean and water masses near Antarctica are ice-free in the 6xCO2 PETM scenario, linked to elevated sea surface temperatures and the associated ice-albedo feedback as well as changes in ocean circulation.



### 3.4 The Sensitivity of Early Eocene Climate to Greenhouse Gas Forcing

The CESM1.2 equilibrium climate system sensitivity (ECS), defined as the ratio of the equilibrium temperature change to the
doubling of the atmospheric $CO_2$, is 2.8 °C under Eocene boundary conditions (Fig. 10a), as compared to 3.2 °C for present-
day conditions (Bitz et al., 2011). A climate sensitivity of 2.8 °C is comparable to the ECS in previous climate simulations
using the climate community system model CCSM3 (Winguth et al., 2010; Lunt et al., 2012; Carmichael et al., 2016; Lunt et
al. 2021), which range from 2.1°C to 4.1°C in the scenario with low cloud albedo feedback. Predicted global mean surface
temperature for the 1x$CO_2$ scenario under Paleocene/Eocene boundary conditions exceeds the value inferred from the
preindustrial scenario with CESM1.2 due to changes in geography, ice sheet-induced climate, and soil albedo forces.
Differences between this study and early Eocene CCSM3 simulations (Winguth et al., 2010; Huber and Caballero, 2011; Kiehl
and Shields, 2013) can be attributed to higher climate sensitivity of CESM1.2, an improved radiative transfer model, and
differences in the soil albedo and aerosol forcing. Note that the CCSM3 prescribed present-day aerosol forcing including the
anthropogenic perturbation, whereas this version of CESM1.2 applied the aerosol forcing of Herold et al. (2014), as estimated
for the early Eocene. Discrepancies between this version CESM1.2 with Community Atmosphere Model Version 4 (CAM4)
and CAM5 (Zhu et al., 2019; see also Fig. 1 in Lunt et al., 2021) are attributed to differences in cloud parameterizations and
radiative transfer (Kay et al., 2012), surface albedo, and prescribed aerosol emissions resulting in higher climate sensitivity in
CESM1.2 with CAM5.

### 4 Proxy- Model Comparisons

### 4.1 Surface Temperature

A suitable range of proxy data exists both for surface temperatures and precipitation. Data sets for temperature have been
compiled as part of the DeepMIP by, e.g., Hollis et al. (2019) and references therein, for three selected time intervals: the LP,
PETM, and early Eocene climatic optimum (EECO). Sea surface temperatures for Paleocene and Eocene have been compiled
from oxygen isotopes, Mg/Ca values, clumped oxygen isotope values in well-preserved foraminifera in clay-rich sediments,
and from Tex 86 in bulk sediment (Pearson et al., 2001; Sexton et al., 2006, Auderset et al., 2022). More details on the data and
uncertainty as used in this study is given by Hollis et al. (2018) and literature cited therein.

The surface temperature is compared for two simulations, the 3x $CO_2$ scenario equivalent to latest Paleocene (LP) conditions,
and the 6x $CO_2$ scenario representing the PETM, as suggested by DeepMIP protocol (Lunt et al., 2017). Comparison of
simulated sea surface temperatures from the 3x $CO_2$ LP scenario with reconstructed paleo proxies on the location of the data
(Fig. 11a) from the latest Paleocene reveals a strong positive correlation (r= 0.748, p =0.66) for mean model values about 2.1
°C cooler at the proxy locations, primarily due to the high latitude bias. This p value is statically not significant because of the
small sample size. The correlation of sea surface temperature between the 6x $CO_2$ PETM scenario (Fig. 11b) and corresponding
PETM data of Hollis et al. (2019) (r= 0.754, p =0.35) is also scattered, despite higher data size. Model-data biases of up to 8°C



and more occur at high latitudes, in particular at the data locations in the Arctic Ocean and New Zealand, potentially linked to
seasonal and/or diagenetic biases (Hollis et al., 2018, Lunt et al. 2021).

## 4.2 Precipitation

Mean annual precipitation (MAP) estimates from sedimentary proxy reconstructions and model simulations at the proxy sites
for the 3xCO$_2$ PAL LP and 6xCO$_2$ PAL PETM scenarios are shown in Figure 12. Proxy sedimentary reconstructions for the lt
Paleocene, PETM, and early Eocene have been described in Carmichael et al. (2016) (their Table 3S). These proxies have been
compiled from oxygen isotopes from mammalian, fish, and foraminiferal fossils (Zachos et al., 2006; Zacke et al., 2009;
Clementz and Sewall, 2011), geomorphological data (John et al., 2008; Schmitz and Pujalte, 2007), biomarkers (Handley et
al.,2011; Pagani et al., 2006), other microfossils (Sluijs et al., 2011; Kender et al., 2012) and sedimentary-inferred precipitation
estimates (for example, Huber and Goldner, 2012). Simulated values within the 3x3 grid cells in the proximity of the proxy
locations have been averaged over these locations. The standard error has been compiled from the deviation of the mean from
these considered grid cells. Thus, a stronger gradient among the grid cell leads to a larger standard deviation and therefore a
larger error. The proxy-inferred estimates of precipitation are represented by a horizontal scale much smaller than resolved by
the inferred predictions from the simulations in this study.

In general, simulated values of MAP agree within the uncertainties of the paleo-precipitation estimates, particularly for the
Chicaloon Formation (Fig. 12b), Western Interior of the U.S. (Fig. 12e), Guchengzi Formations (Fig. 12f), Central Europe
(Fig. 12g), Laguna del Hunco (Fig. 12j) and Waipara River (Fig.12k). Tropical precipitation appears to be sensitive to sea
surface temperature (and CO$_2$ radiative forcing), as identified for the Cerrejon Formation, Columbia (Fig. 12h), and also
suggested by analysis of present-day observations (e.g., Good et al. 2021, and references therein).

The model-data bias appears to be largest in the intertropical convergence zone (ITCZ, Fig. 12h) as well as at southern polar
latitudes (Fig. 12g), with a too low simulated precipitation compared to the observations, whereas CESM1.2 simulations
overestimated precipitation (as in previous studies, e.g., summarized by Carmichael et al., 2016). Note that data records span
wide time intervals from the late Paleocene to the Eocene Climate Optimum. Notably, paleo-precipitation estimates from
sedimentary records for the Antarctic Peninsula (see Figure 12 g) appear to be highly variable (with a range of ~70 cm/yr to
~260 cm/yr; Carmichael et al., 2016; their Table 3S). Uncertainties exist also about seasonal biases for paleo-precipitation.
Present-day simulated strong and very strong precipitation events in CESM1.2 are well reproduced, but model-data biases may
be associated with the simulated double ITCZ as indicated by Gent et al. (2011). This error also appears in the Eocene climate
simulations.

## 4.3 Deep Sea Ocean Circulation

In the following, simulated abyssal ocean circulation results will be evaluated with deep water mass tracers from the
sedimentary record. The distribution of stable carbon isotope ratios (δ $^{13}$C) of dissolved inorganic carbon (DIC) in the ocean is
dominated by two effects, the fractionation by the air-sea gas exchange due to the tendency of the lighter isotope $^{12}$C to more





easily evaporate in dependence on the temperature (Mook et al., 1974; Broecker and Maier-Reimer et al., 1992), and the biological pump with δ $^{13}$C of DIC being correlated to phosphate below 1000 m (Kroopnick,1985; Östlund et al., 1987, Winguth et al., 1999). However, this correlation may have been affected by a different fractionation of particulate organic matter in the past (Rau et al., 1991). An increased absence of water masses from the surface typically leads to an enrichment

of $^{12}$C and nutrients due to remineralization of particulate matter, thus deep present Pacific water masses are characterized by high phosphate and low δ $^{13}$C values. Analysis of stratigraphic reconstruction of δ $^{13}$C in planktonic and benthic foraminifera during the Early Eocene (Sexton et al. 2006) suggests that a lower-than-present vertical δ $^{13}$C gradient resulted from enhanced vertical mixing and thus a lower age in water masses in the abyss. Deep sea flow patterns inferred from δ $^{13}$C gradients indicate strong water mass formation around Antarctica including the South Atlantic (present Weddell Sea) and South Pacific (near the

Eocene Australia) prior to the PETM and their decline during the PETM. Northern component water mass formations remained in the North Atlantic Ocean during the PETM (Nunes and Norris, 2006; Kirtland Turner et al., 2024) thus leading to a reversal of the deep-sea circulation in the Atlantic Ocean. Modelling results with CESM1.2 in this study support deep water formation in the South Atlantic (in the proximity of the present Weddell Sea) under Latest Paleocene conditions (3x CO$_2$ LP experiment) and a diminished Southern Ocean component water mass during the PETM (6x CO$_2$ PETM experiment). The decline of

southern component water mass formation for the 6x CO$_2$ PETM experiment (Figure 13) is in agreement with the meridional trends inferred from the distribution of CaCO$_3$ dissolution horizons (Penman and Zachos, 2018; Zeebe and Zachos, 2007) and chert sedimentary records (Penman et al., 2018). It is noteworthy that the South Pacific deep-water formation is only simulated in the 1x CO$_2$ PD experiment (Figure 13a).

Neodymium isotope ratios ($^{143}$Nd/$^{144}$Nd expressed as ε$_{Nd}$; DePaolo and Wasserburg, 1976) from fish teeth have been commonly

applied as a deep water mass tracer (Tachikawa et al., 1999; Martin and Scher, 2004; Thomas et al., 2003; Thomas 2004; Via and Thomas; 2006). Nd isotope data from sedimentary records of the Pacific Ocean are supportive of convective overturning in both polar hemispheres for the early Cenozoic (Thomas et al., 2014, McKinley et al., 2019). This study supports a separation of the sources of deep water in the South Pacific Ocean from those in the Atlantic Ocean but does not support polar deep-water source formation in the North Pacific Ocean (Figure 13 a-c). However, an inflow from the Central American Sea from the

deep Atlantic to the Pacific Ocean could have ventilated the abyssal Pacific Ocean. Deep water formation in the North Pacific Ocean remains uncertain based on this study as well as on previous DeepMIP studies (Zhang et al., 2021). Data coverage of Nd isotopes for the Atlantic Ocean is limited and their analysis may favor bipolar deep-water formation. However, other studies suggest that deep water formation in the North Atlantic Ocean did not occur prior the middle Eocene (e.g., Boyle et al., 2017; Coxall et al., 2018; Gleason et al., 2009; Hohbein et al., 2012).

## 5 Summary and Conclusions


Climate sensitivity in this CESM1.2 simulation with CAM4 is comparable to previous CCSM3 simulations, but this study highlights several improvements in representing the climate change across the Paleocene-Eocene boundary and more



reasonably represents the observations. This is particularly the case for the prediction of temperatures in both the north and the south polar regions, and the substantially reduced pole-to-equator temperature gradient. The CESM1.2 also produces a better

representation of tropical precipitation but underrepresents the proxy-inferred magnitude. This may be partially linked to the model resolution, but also to simulated warmer climate enhancing the water vapor saturation value in the atmosphere.

A shift from a southern-dominated deep-sea ventilation (thermal mode) at low atmospheric $CO_2$ radiative forcing of 1-3x$CO_2$ PAL to a less vigorous intermediate-to-shallow ventilation (haline mode) at atmospheric $pCO_2$ equal to or exceeding 6x$CO_2$ PAL has been simulated. A simulated decline in the southern component relative to the northern component water mass

formation in response to $CO_2$ radiative forcing during the onset of the PETM is consistent with carbon isotopes, $CaCO_3$ dissolution horizons, and change in the chert sedimentary record. However, a bipolar deep-sea ventilation as suggested from Nd isotope ratios cannot supported by this as well as other DeepMIP studies (Zhang et al. 2021). The lack of Antarctic ice during the late Paleocene/early Eocene and associated reduced pole-to-equator temperature gradient affects the strength of wind stress and thus the strength of the barotropic stream function, enhancing the stratification close to Antarctica.

Reduced Ekman-induced wind stress over the tropics contributes to reduced upwelling and thus favors a reduced paleo productivity in that region. Decline in deep-sea ventilation and oxygen solubility due to ocean warming would have caused a decline of dissolved oxygen concentrations in the abyssal ocean (Nicolo et al., 2010; Schmidko et al., 2017; Ito et al., 2017; Winguth et al., 2013) and enhanced benthic extinctions during the onset of the PETM. However, dissolved oxygen concentration in the upper tropical ocean may have increased during the PETM at least regionally due to decreased upwelling

and productivity (Moretti et al., 2014). This greater response in ocean stratification due to an increase in $CO_2$ radiative forcing is linked to higher climate sensitivity and improved cloud feedbacks of CESM1.2 compared to CCSM3.

**Data Availability**

Data will be provided by the corresponding author upon request.

**Author contributions**

AW performed climate simulations and analysis, and CS provided boundary conditions including aerosol forcing and land surface maps for the simulations. MB did postprocessing and averaging of climate simulation and analysis for the 1x$CO_2$ PB scenario. PH, ET, AW, and CW discussed the results. AW wrote the paper, and all authors provided input for the paper before submission.



**Competing Interest**

Arne M.E. Winguth is a member of the editorial board of the journal "Climate of the Past". The other authors declare that they have no conflict of interest.

**Acknowledgements.** This research used samples and data provided by the International Ocean Discovery Program (IODP). Funding for this research was provided by National Science Foundation (NSF) grants OCE-1536630 to A. Winguth and E. Griffith and OCE-1536611 to E. Thomas and by Heising-Simons Foundation Grant #2016-004. We acknowledge high-

performance computing support from the Cheyenne: HPE/SGI ICE XA System (doi:10.5065/D6RX99HX) and from the Casper system (https://ncar.pub/casper) provided by the NSF National Center for Atmospheric Research (NCAR), sponsored by the National Science Foundation.



**Table 1.** List of experiments for the climate simulations

| Experiment | | pCO$_2$ [ppmv] | Aerosol forcing | SAT [°C] | TS Land [°C] | TOA[2] [W m$^{-2}$] |
|---|---|---|---|---|---|---|
| Preindustrial (PI) | | 280 | 1850 | 14.8 | 8.0 | -0.555 |
| 1xCO$_2$ | Paleocene Baseline (PB) | 280 | PETM[1] | 22.2 | 16.7 | -0.554 |
| 3xCO$_2$ | Latest Paleocene (LP) | 840 | PETM | 26.0 | 22.1 | -0.290 |
| 6xCO$_2$ PETM | | 1680 | PETM | 28.9 | 26.4 | 0.023 |
| 12xCO$_2$ PETM | | 3360 | PETM | 33.0 | 32.5 | 0.270 |

[1] see Herold et al. (2014); [2] top of atmosphere energy balance

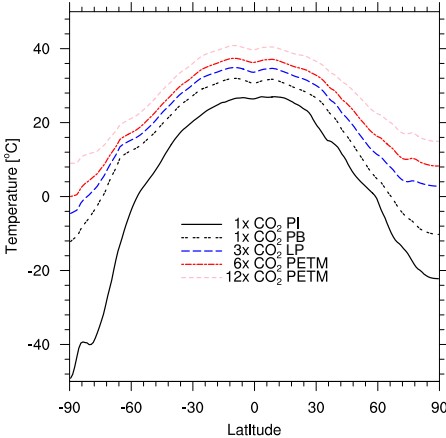

**Figure 1.** Zonal average mean annual surface air temperature simulated by CESM1.2 for the preindustrial experiment (PI;
black), Paleocene Baseline (PB) simulation at 1xCO$_2$ PAL (double dashed black), latest Paleocene (LP) at 3xCO$_2$ PAL (dashed
blue), PETM 6xCO$_2$ PAL (red, dash-dotted), and PETM 12xCO$_2$ PAL (light red, dash) experiment.



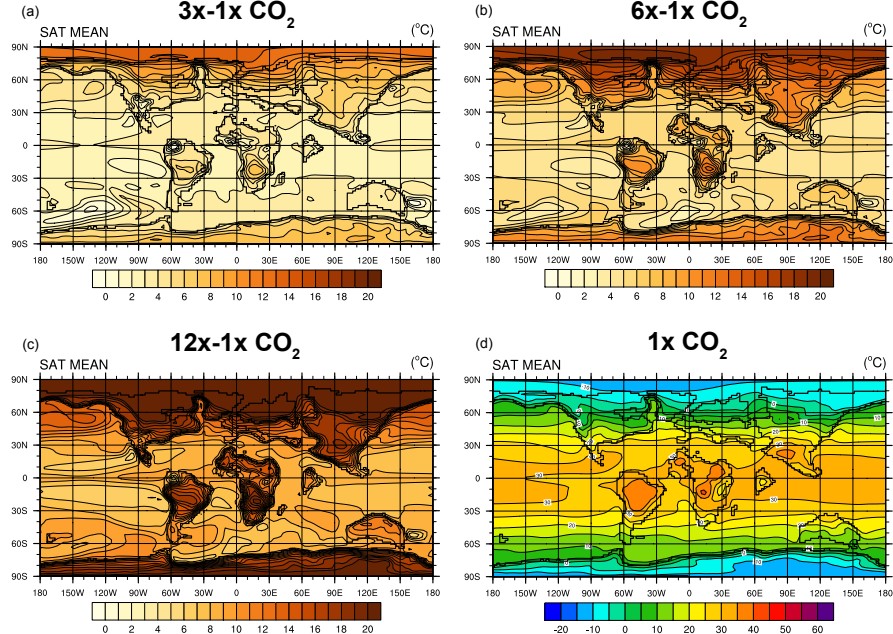

**Figure 2.** Difference in mean annual 2m surface air temperature in °C simulated by CESM1.2 between (a) LP at $3xCO_2$ PAL, (b) PETM at $6\ xCO_2$ PAL, and (c) PETM at $12\ xCO_2$ PAL and PB at $1xCO_2$ PAL experiment. (d) PB at $1xCO_2$ PAL scenario is shown for reference.



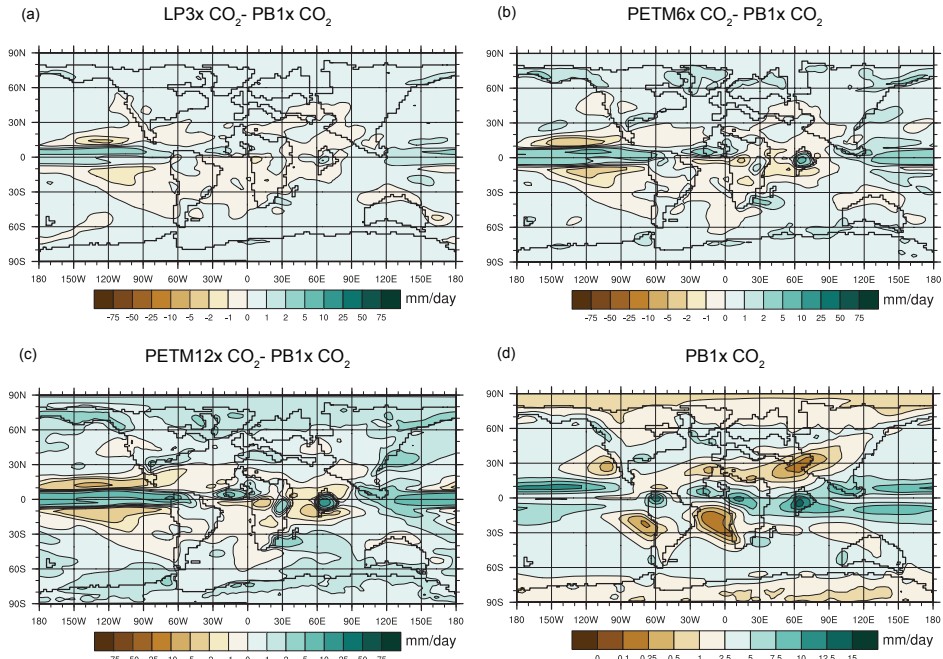

**Figure 3.** Difference in mean annual precipitation in mm day$^{-1}$ simulated by CESM1.2 between (a) LP at 3xCO$_2$ PAL, (b) PETM at 6 xCO$_2$ PAL, and (c) PETM at 12 xCO$_2$ PAL and PB at 1xCO$_2$ PAL experiment. PB at 1xCO$_2$ PAL scenario is shown for reference.




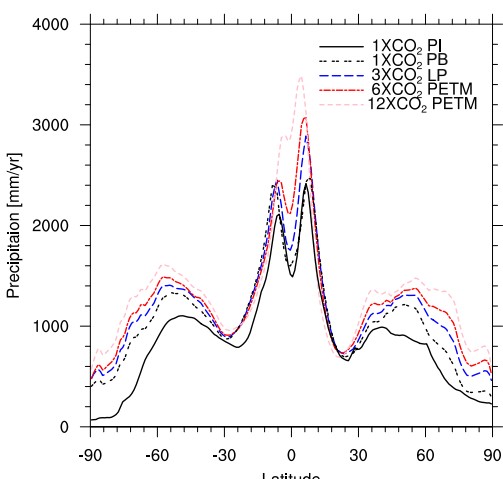

**Figure 4.** Zonal average 2 m surface mean annual precipitation simulated by CESM1.2 for the preindustrial experiment (PI; black), Paleocene Baseline (PB) simulation at 1xCO$_2$ PAL (double dashed black), Latest Paleocene (LP) at 3xCO$_2$ PAL (dashed blue), PETM 6xCO$_2$ PAL (red, dash-dotted), and PETM 12xCO$_2$ PAL (light red, dash) experiment.


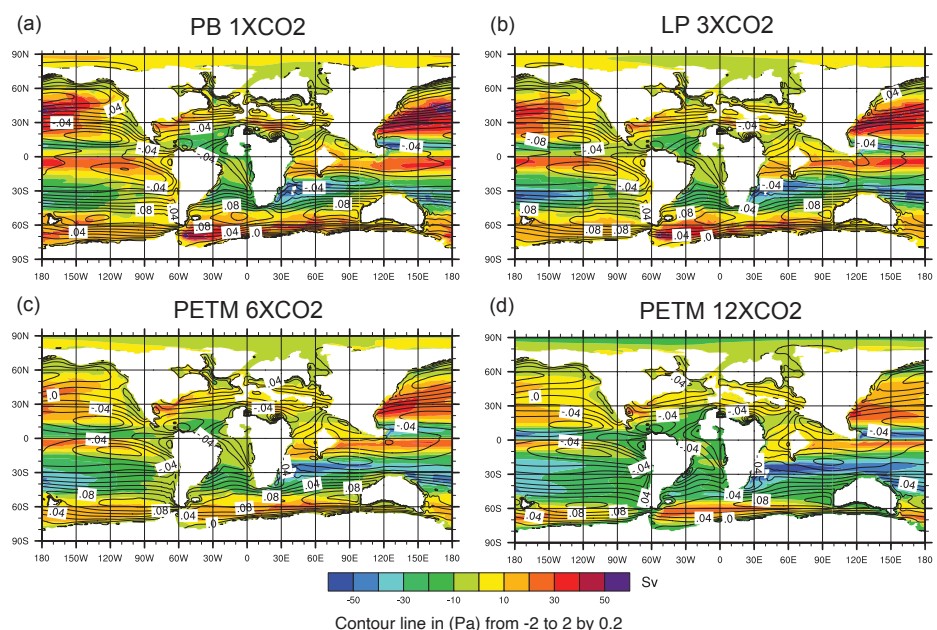



**Figure 5.** Mean annual barotropic stream function in Sv (1 Sv = $10^6$ $m^3$ $s^{-1}$; contour shades) and zonal wind stress in Pa (contour lines) simulated by CESM1.2 between (a) PB at 1xCO$_2$ PAL, (b) LP at 3xCO$_2$ PAL, (c) PETM at 6 xCO$_2$ PAL, and (d) PETM at 12 xCO$_2$ PAL experiment.

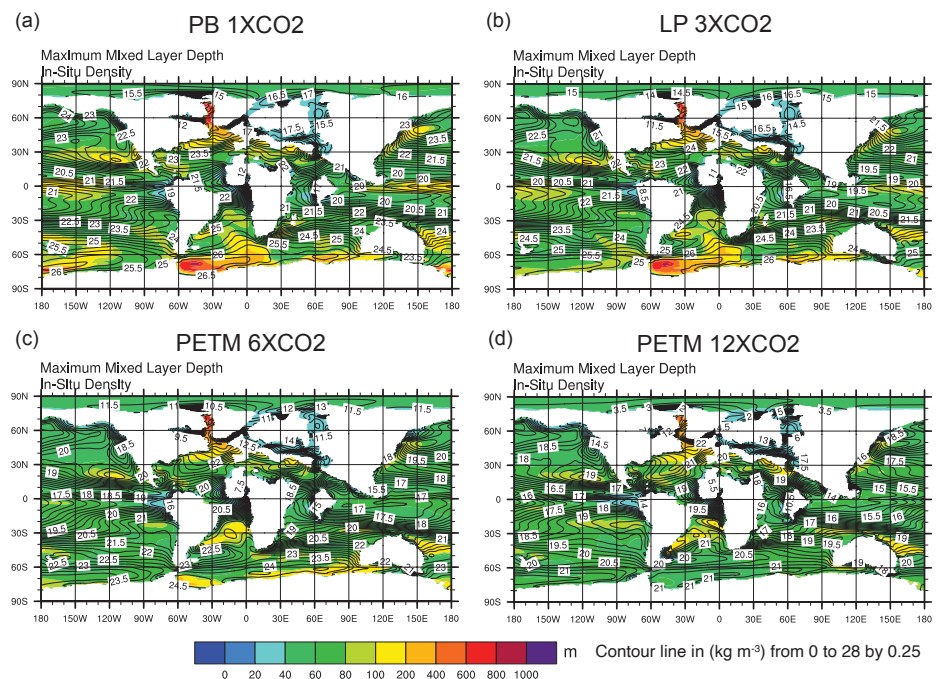

**Figure 6.** Mean maximum mixed layer depth in m (contour shades) and In-Situ Density $\sigma_t$ in kg $m^{-3}$ (contour lines) simulated by CESM1.2 between (a) PB at 1xCO$_2$ PAL, (b) LP at 3xCO$_2$ PAL, (c) PETM at 6 xCO$_2$ PAL, and (d) PETM at 12 xCO$_2$ PAL experiment.



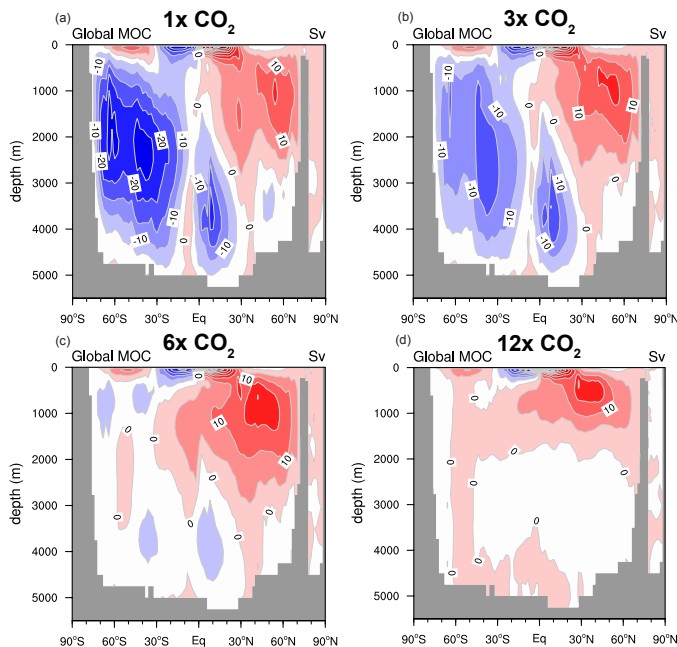

**Figure 7.** Mean annual global meridional barotropic stream function in Sv simulated by CESM1.2 between (a) PB at $1xCO_2$ PAL, (b) LP at $3xCO_2$ PAL, (c) PETM at $6 xCO_2$ PAL, and (d) PETM at $12 xCO_2$ PAL experiment.



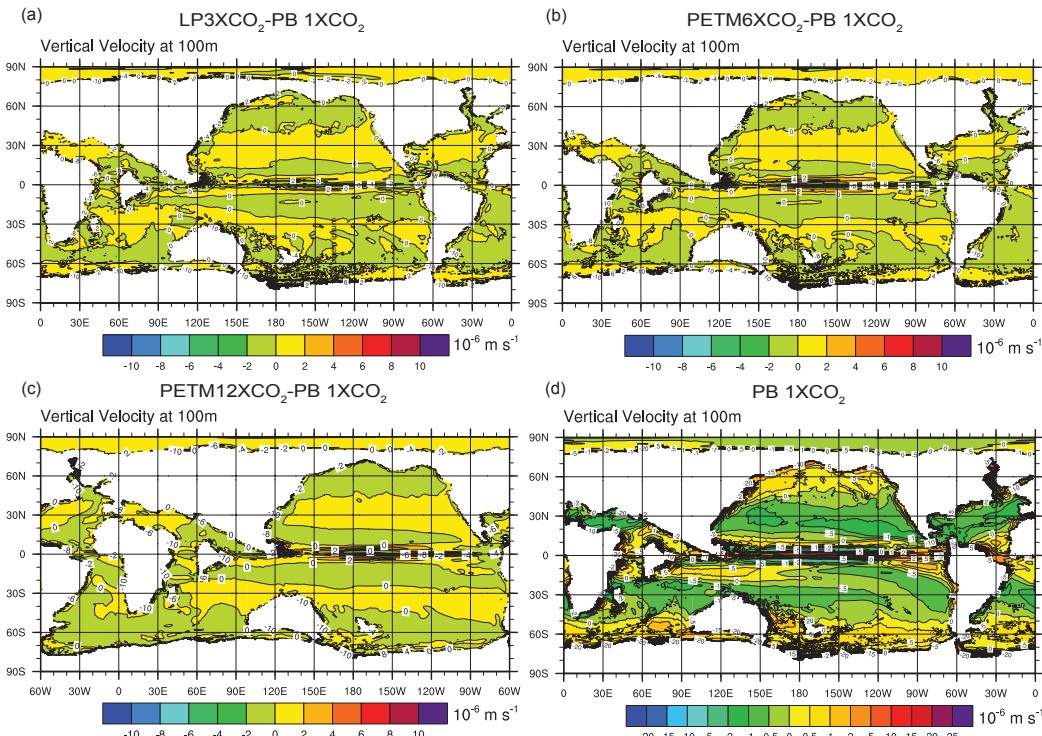

**Figure 8.** Mean annual global upwelling in 100 m in $10^{-6}$ m s$^{-1}$ simulated by CESM1.2. Difference between (a) LP at 3xCO$_2$ PAL and PB at 1xCO$_2$ PAL scenarios, (b) PETM at 6 xCO$_2$ PAL and PB at 1xCO$_2$ PAL scenarios, (c) PETM at 12 xCO$_2$ PAL and PB at 1xCO$_2$ PAL scenarios, (d) PB at 1xCO$_2$ PAL.





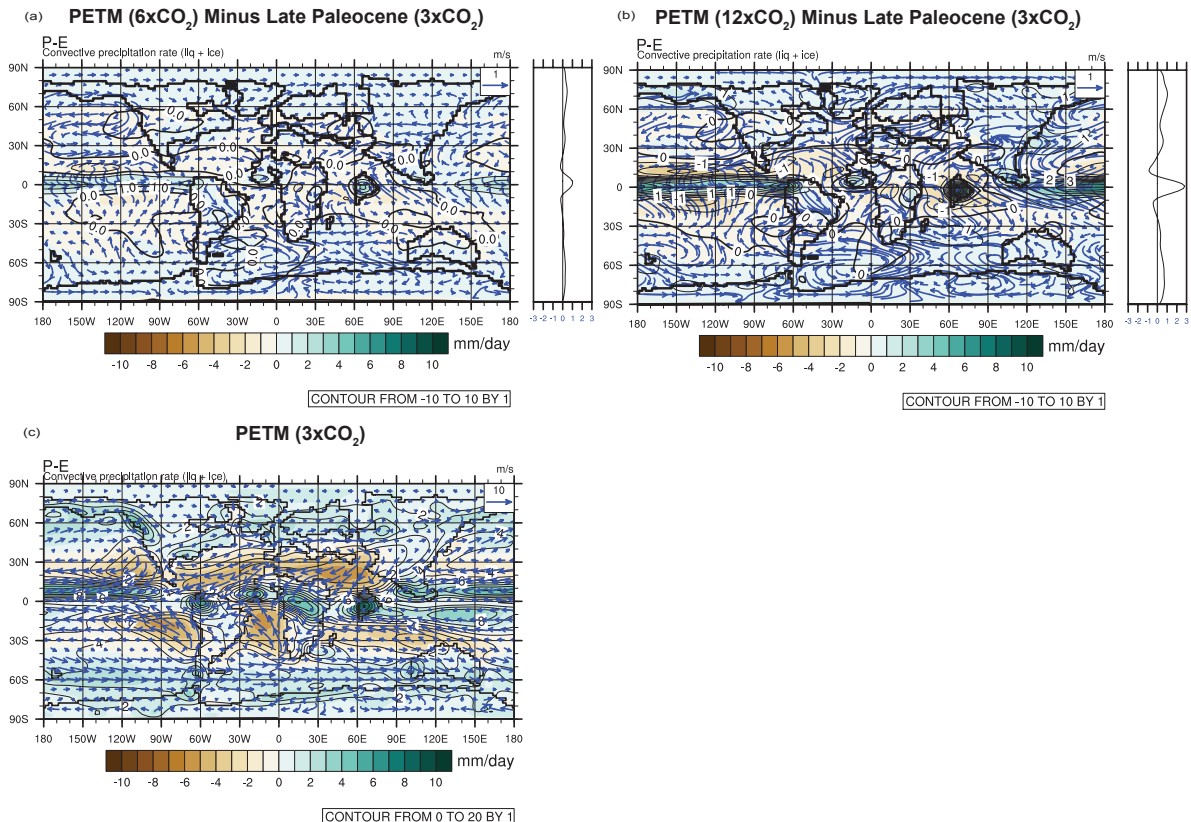

**Figure 9.** Difference in mean annual precipitation minus evaporation (contour shades) in mm day$^{-1}$, precipitation (contour lines) in mm day$^{-1}$, and wind speed (vectors) simulated by CESM1.2 between (a) PETM at 6 xCO$_2$ PAL and LP at 3xCO$_2$ PAL experiment, (b) PETM at 6 xCO$_2$ PAL and LP at 3xCO$_2$ PAL experiment, and (c) LP at 3 xCO$_2$ PAL scenario. Wind speed vector length for (a) and (b) is 1 m s$^{-1}$ and for (c) 10 m s$^{-1}$.



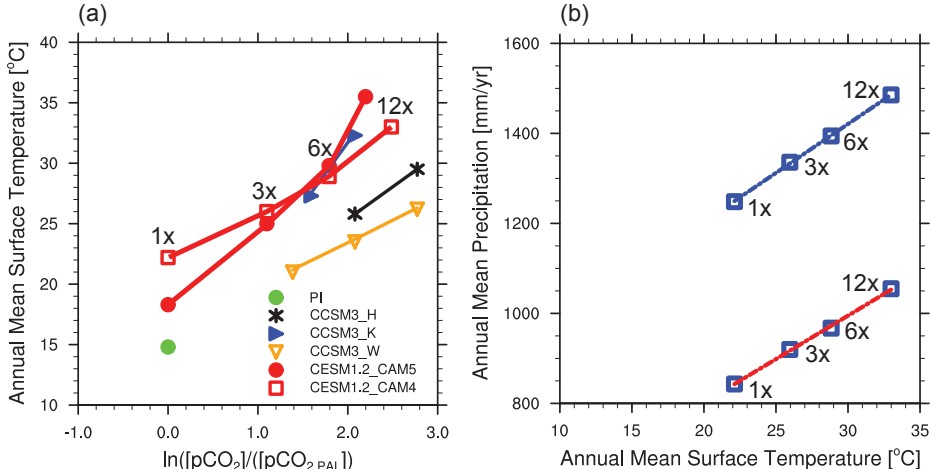

**Figure 10.** (a) Mean annual surface temperature as a function of the natural logarithm of the atmospheric $pCO_2$ relative to the $pCO_2$ at preindustrial levels. For comparison, simulations of CCSM3_W (Winguth et al., 2010), CCSM_H (Huber and Caballero, 2011), CCSM3_K (Kiehl and Shields, 2013), and CESM1.2 with CAM5 (Zhu et al., 2019) are shown. (b) Annual
380   mean precipitation (blue) and convective precipitation (red) for CESM1.2 (this study) in relation to annual mean 2m surface air temperature.

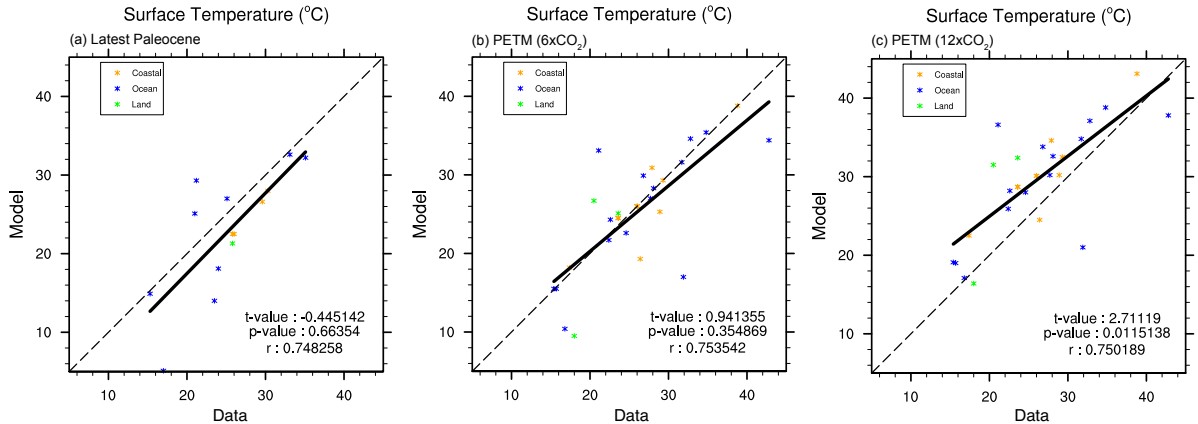

**Figure 11.** Pearson correlation between compiled temperature reconstructions (data from Hollis et al., 2019; orange star
385   coastal, blue star ocean, and green star land sites) (a) for the Latest Paleocene and the CESM1.2 LP at 3xCO2 PAL scenario, (b) for the PETM and the CESM1.2 PETM at 6xCO2 scenario, and (c) for the PETM and the CESM1.2 PETM at 12xCO2 scenario.

.

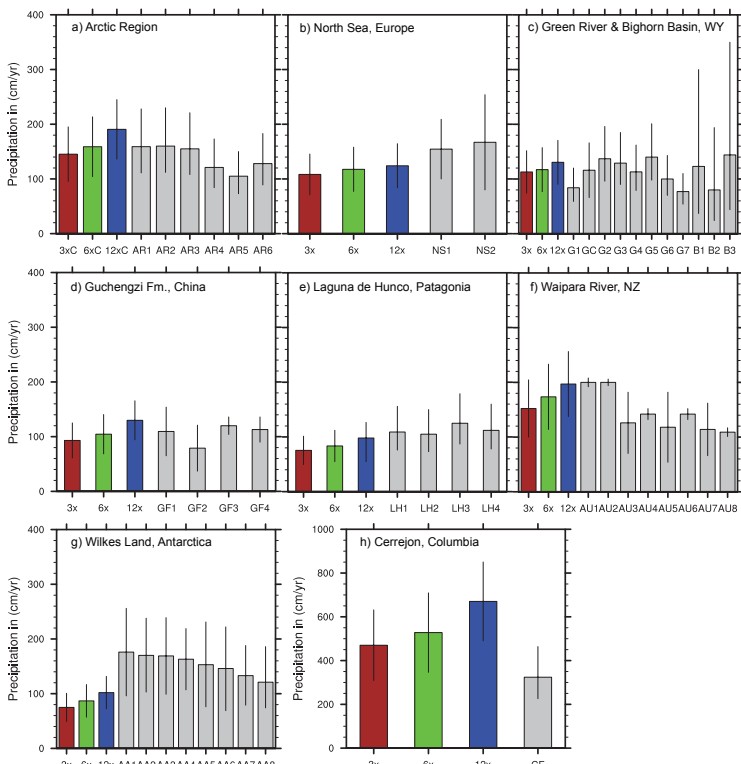

**Figure 12.** Comparison of mean annual precipitation between Late Paleocene/Early Eocene proxy data (grey) and CESM1.2
LP 3x CO₂ (red), PETM 6x CO₂ (green),  and PETM 12x CO₂ (green) experiments at the proxy sites for (a) Arctic region
including Chicaloon Formation, Alaska (AR1-AR4), Arkrose Ridge Formation, Alaska (AR5, Sunderlin et al. 2011), and
Northwest Territory (AR6, Greenwood et al. 2010); (b) North Sea, Europe (NS1-NS2; Eldrett et al., 2014), (c) Green River
Basin (G1-G7; Wilf et al., 1998; and Wilf, 2000; and GC; Wing and Greenwood, 1993) and Big Horn Basin (B1-B3, Wing
et al. 2005) in Wyoming, North America, (d) Guchengzi Formation, Fushun, China,(GF1-GF4; Wang 2010; Wang et al.
2013; and Quan et al., 2011), (e) Laguna de Hunco, S. America (LH1-LH4; Wilf et al., 2005), (f) Waipara River in New
Zealand (NZ1-NZ8; Pancost et al. 2013), and (g) Wilkes Land Sector, Antarctic Peninsula (AA1-AA7; Pross et al. 2012).
Note that the scale needed to be adjusted for (h). For the geographic position of the proxy records and details see text and
Table S3 in Carmichael et al. (2016).




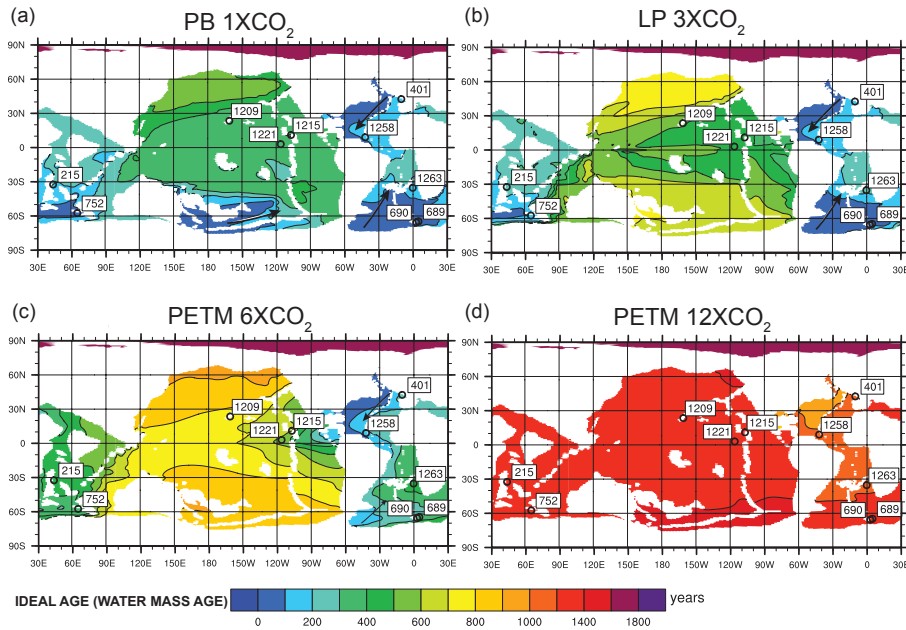

**Figure 13.** Ideal age water mass tracer in years simulated by CESM1.2 between (a) PB at 1xCO$_2$ PAL, (b) LP at 3xCO$_2$ PAL, (c) PETM at 6 xCO$_2$ PAL, and (d) PETM at 12 xCO$_2$ PAL experiment. Arrows denotes flow direction of deep water and circles with numbers refer to location of sedimentary records from IODP deep water mass proxies (see references in section 4.1).

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
