# Peer review of "Collapse of Deep-Sea Circulation during an Eocene Hyperthermal Hothouse – A DeepMIP Study with CESM1.2"

_EGUsphere, 2024_

## Referee Comment (RC1)

Comments on *Collapse of Deep-Sea Circulation during an Eocene Hyperthermal Hothouse – A DeepMIP Study with CESM1.2*
By *Winguth et al.*

**General comments:**

- I would like to see more information on the spin-up and equilibration of the simulations, as this can be of key importance to interpret the results.
- Generally, the manuscript needs a thorough check for readability and language; many sentences are rather tedious to understand and contain (minor) inconsistencies or errors.
- What about seasonal responses? e.g. are changes in precipitation linked to monsoons/storm tracks/ITCZ shifts?
- Most of the figures shown are rather generic, much of the patterns and responses shown are hardly discussed while many of the processes mentioned are not supported by the fields considered (e.g. circulation, radiation, ice, clouds).
- In much of the results section, the design of the figures does not really support the overall structure, e.g. panels a-c of figures 3/4 are discussed in sections 3.2/3.3, while panel d is discussed in 3.1.
- I am missing a clear role of radiative feedbacks related to clouds in much of section 3, while this is mentioned up front as a key contribution.
- In the results section, I am missing an assessment of circulation changes, which are at times mentioned but not shown or referred to.
  edit: I see near surface winds in figure 9, but there is little reference to this figure and the figure needs some improvement for readability.
- A much more careful consideration of the 12xCO2 (and 6xCO2?) simulation is in order, particularly considering the consequences for ECS and overturning. Despite 2000 years of simulation, the final state may still be highly dependent on the initial conditions implying very warm low-latitude intermediate waters.
- The conclusion/discussion section is very limited, with the majority focussing on DWF and ocean ventilation and thus not representing the main findings well. Although this is in the main title, I found limited information in the results that support the suggested collapse of the deep-sea circulation.

**Specific comments**:

- The abstract needs some improvement, as it is rather tedious to read and several comparison statements lack a clear reference.
- L15: what does PAL stand for?
  I see this is clarified on L72, but I would argue the term 'levels' is a bit ambiguous? This may suggest you are increasing non-CO2 components as well. If this is following general syntax used in other work, ignore this comment.
- L16: 'equatorial warming' is a bit ill-defined here. In addition, it is unclear to interpret the 36.9C value without a reference from proxies or e.g. 1x/3xCO2.
- L62: Is this adjustment shown somewhere? Even after 2000 years, equilibration may be highly dependent on e.g. initial conditions.

- L85: The initial temperature profile leads to some questions;
    - If this is used for all experiments, they likely have very different levels of, and potential biases from equilibration, is this accounted for?
    - The linear temperature profile with depth should lead to some very warm intermediate levels, which are physically unrealistic and may lead to large temperature biases lasting many 1000s years in stable regions.
    e.g. temperatures at 1000-2000m depth at low latitude are generally 10-15C in similar 3/4xCO2 Eocene simulations, while they would be ~32-36C initially. I would not expect such a large deviation to dissipate in these simulations.
    - Is there a reason to have a discontinuity at z=5000m and scale with 6000m instead?
- L89: In contrast to the very warm initialisation, there is a positive net TOA radiative balance (I assume it is defined positive downward?). While still quite large, I would expect negative values considering the initialisation. Some further analysis of the time evolution of e.g. depth-dependent temperature and radiative balance would be helpful here.
- L103: Why would this effect be uniquely important to West-Antarctica, rather than all high-latitude regions?
- L109: This is confusing; global precipitation is linked to polar temperatures rather than warmer SSTs overall? It is also not straightforward to me how land albedo is linked directly to precipitation.
- L113: Where do we see precipitation differences over land relative to the PI scenario? The argumentation is a bit weak here; why would a strengthened ITCZ have a different effect over land versus ocean?
- L120: I am missing an explanation on the drying patterns close to the equator over much of the ocean. These do not align with the downward branches of the Hadley cells, so are seemingly related with a migration and/or contraction of the ITCZ which further intensifies at higher CO2? (edit: I see this is the topic of sections 3.2/3.3, but still miss a clear explanation there).
- L129: This is mostly a repetition of an earlier paragraph. Also, 200Sv is mentioned here versus >180 above?
- L145: I do not see SST in Table 1? Is the value the same by coincidence, or a typo?
- L149: Is the ice albedo feedback shown somewhere? As sea ice is said to only be present in winter, this would unlikely have a major radiative effect.
- L168: I assume this is about the global average?
- L171: The decline in latent heat flux is contradictory to what was said earlier (and further down; L180) when considering rainfall, if this is land-only versus global some clarification is needed. More generally, a discussion on the radiative balance would be helpful (which I believe is shown in the general DeepMIP paper including these simulations?). What about the potential role of albedo and clouds? Are the land temperature changes highly seasonally dependant?
- L172: This is again land only?

- L173: It is a bit counterintuitive in this section to generally treat the 3xCO2 case as a reference, while this is not shown as such in the figures. While it does make sense following up on the previous section, this is not consistent with the figures and makes the interpretation more difficult. For example: further strengthening of equatorial rains is seen for 6x/12x CO2, consistent with the 3xCO2 anomaly pattern, but there is a clear double ITCZ at 1xCO2. It is therefore difficult to assess whether the former show only a further strengthening of the anomaly pattern, or the background as well.
- L176: Where/how can I see the enhanced moisture transport mentioned here?
- L179: An increased downward branch of the Hadley cell is mentioned (but not shown?) explaining drier conditions in the subtropics. Although reasonable, I think the assessment of precipitation patterns is too limited here;
  - Overall tropical rains are increasing towards higher T/CO2, but the rain bands are contracting towards the equator.
  - Tropical rainbands cover a significantly wider latitude range in the PB versus PI, but contract again towards higher CO2. Can you explain this?
  - While evaporation is higher due to lower saturation in a warmer atmosphere, precipitation does not decrease in the subtropics for higher CO2.
  - Is there a significant role of monsoonal rains in understanding changes in precipitation?
  - Extratropical rains increase significantly and expand polewards for higher CO2, while the meridional temperature gradients reduce. What does this mean for midlatitude storm tracks?
- L181: I assume we are looking at figure 5 here? In that case, jumping from zonal wind stress on the ocean to trade winds is a bit steep. In its current form, the wind stresses are also quite tough to assess from the figure.
- L182: As a reader, it is tough to assess the role of the mentioned albed-related feedbacks as they are not shown or referred to.
- L183: Looking at Figure 8, Ekman upwelling seems to be displaced as much as being reduced? This is, however, hard to see clearly from the figure.
- L187: conclusions are made here that deserve more careful consideration; without a thorough assessment of deep ocean equilibration it is near impossible to have a clear conclusion on the overturning circulation state. Looking at the age tracers in figure 13 (which are very relevant, but not mentioned up to this point?), my suspicion is that much of the deep ocean is completely stagnant and therefore dominated by the initial conditions. As the latter are very warm, this run likely has a TOA imbalance which appears acceptable, but may not at all mean that the ocean is adjusted to the applied forcing. Despite relatively weak and shallow, the northern overturning cell at 12xCO2 still indicates sinking mostly at 40-60N and down to 1-1.5km. I do not see how this represents a subtropical haline mode?
- L191: Again, I miss a clear assessment of the actual ice-related feedbacks here. I assume the 3xCO2 case only has ice in the wintertime, so any ice albedo-related feedback should be minimal.

- L195: Is this value adjusted for possible transient forcing? With 2000 years of simulation, this should be feasible (see e.g. Baatsen et al 2020; Figures 2 and 10). The reported value is lower than the one we found for our 38Ma simulations (3.2C) using CAM4, which is expected to be lower compared to CAM5. While the Eocene simulations in Baatsen et al. (2020) were reasonably well equilibrated after ~4000 years (see e.g. Figures 2 and S2), a more significant deviation was found in our shorter (1000-2000year) Pliocene simulations (Baatsen et al 2022), but these values could be easily corrected using the Gregory et al extrapolation method (Figures 2,3 and S2-S4).
- L199: It is unclear to me which inferred value is considered here.
- L205: It is new to me at this point that both the CAM4/CAM5 configuration was used for this study, please clarify! In the previous results, which configuration is considered?
- L220: Although intuitive, a direct comparison between proxy and model temperatures is not very informative (but I still appreciate the scatter plots shown). For example, a simulation overestimating tropical temperatures by 10C and underestimating polar ones by the same amount could still yield a perfect correlation. In addition, the meridional temperature gradient is such a basic pattern that any model should show a reasonable correlation. If anything, consider RMS error values or correlate temperature anomalies relative to PD/PI values.
- L225: What are the actual findings here?
- L236: There is a disclamer here on the scale used for error margins in the model results, but I am missing a clear motivation of the latter. Also, when using simple grid cells (which should be fine), it is good to notice the differences in scales due to the model grid.
- L244: This is confusing: is the model precipitation too low, or too high compared to proxies? Are we not considering CESM1.2 here?
- L245: Can anything be said about precipitation changes between LP and PETM? This is argued to be the main motivation for the different $CO_2$ forcings.
- L250: Although this is a valid point, there is no clear sign (as far as I am aware) that a double ITCZ in the Early Eocene would not have been present.
- L253-267: As this is purely a discussion of prior knowledge, this part should be either in the introduction or discussion? The same holds for L270 onwards, making this mostly a discussion in addition to 2-3 lines presenting results.
- L270: Extreme care is needed here, considering the likely limited adjustment of deep ocean water masses in these simulation.
- L291: I do not fully understand how enhanced WV saturation would limit precipitation
- L293: I am not convinced about this, based on the results shown

**Technical remarks:**

- L13 't' redundant?
- L15 and following: use {\times} for 3x and similar? Throughout, syntax is not consistent; e.g. lines 175-185 are very inconsistent in usage of spacing.
- L20: mid-latitude regions?
- L60: To my knowledge 'Finite volume' refers to the dynamical core, which is connected to the horizontal grid but does not specifically define it. Is the ocean grid the 'standard' ~382x320 curvilinear bipolar configuration? Does it also incorporate equatorial stretching?
- L84: Is there a specific use to the underlining of phi?
- L85: 6000z/6000: typo?
- L113: 'the precipitation increases … is linked to'
- L129: capitalise Southern Hemisphere?
- L148: 8C-13C is a bit inconsistent with the use of differences before.
- L153: Colombia or Columbia?
- L157: in the in
- L168: respectively at the end of the sentence?
- L169: $CO_{2}$
- L205: this version of the CESM1.2
- L215: $TEX_{86}$
- L223: up to and more?
- L228: lt = late?
- L288: I would always make a clear distinction between observations and proxy estimates.

**Tables/figures**

- Figure 1: surface or near-surface/sm air temperature?
- Figure 2: As the absolute (zonal average) temperatures are already shown in Figure 1, I feel like it would be more informative to show the response per CO2 doubling here, or otherwise between successive simulations?
- Figure 3: I was confused by the use of the same colormap for anomalies and absolute values, assuming panel d was showing the difference relative to PI.
- Figure 4: This could be combined with figure 1?
  Especially for precipitation, consider using cos(lat) on the horizontal axis such that shifts in precipitation are conserving surface area.
- Figure 5: The contour lines are difficult to interpret here.
- Figure 8: Despite one brief mentioning of reduced upwelling, the relevance of this figure is limited to me. In addition, the use of colour map and scale is far from optimal
- Figure 9: While I appreciate the information shown, this figure is very tough to read due to the overlay of contours and quivers. Using the same colourmap for both absolute values and anomalies is again confusing, particularly as the background field is already symmetric about 0. Additionally, I hardly see any anomalies due to the scaling in panels a,b. Clarification of the zonal averages on the side seems to be missing
- Figures 10-12: the formatting of these figures is very different from other figures, please improve for consistency especially in terms of colours, fonts, and markers used.
- Figure 12: Is there a clear motivation for the use of cm/yr rather than mm/yr or mm/day such as in other figures? A simple map showing these proxy sites overlaid with model precipitation would be a useful addition here.
- Figure 13: I cannot find any mentioning of the depth that is considered here. A clear reference to the precise simulation length of the different experiments is also quite relevant to assess this figure.

---

## Author Comment (AC2)

Reviewer comments in Red and response to reviewer in black

We thank the reviewer for the detailed review and will address the reviewer's comments as listed below and in the revision.

• I would like to see more information on the spin-up and equilibration of the simulations, as this can be of key importance to interpret the results.

In the revision, we have added the time series of temperature and compute temperature drift for the spin-up to discuss the adjustment of the simulations to the boundary conditions.

• Generally, the manuscript needs a thorough check for readability and language; many sentences are rather tedious to understand and contain (minor) inconsistencies or errors.

We have revised the manuscript to make it more readable and consistent.

See above

• What about seasonal responses? e.g. are changes in precipitation linked to monsoons/storm tracks/ITCZ shifts?

The focus of this paper is on the deep-sea circulation and the forcing boundary conditions creating these changes. Thus, addressing seasonality is beyond the scope of this paper and including it in the paper revision would make this paper less focused.

• Most of the figures shown are rather generic, much of the patterns and responses shown are hardly discussed while many of the processes mentioned are not supported by the fields considered (e.g. circulation, radiation, ice, clouds).

We have improved the explanation of figures in the revised manuscript .

• In much of the results section, the design of the figures does not really support the overall structure, e.g. panels a-c of figures 3/4 are discussed in sections 3.2/3.3, while panel d is discussed in 3.1.

In the revision, we addressed this concern and improved the  analysis to be more consistent to the order of the figures.

• I am missing a clear role of radiative feedbacks related to clouds in much of section 3, while this is mentioned up front as a key contribution.

In the revision we have de-emphasized the cloud feedback as part of this study and refer to the already published literature.

• In the results section, I am missing an assessment of circulation changes, which are at times mentioned but not shown or referred to.

We have added an assessment of the circulation changes in response to changes in the atmospheric $CO_2$ radiative forcing in the revision of this manuscript.

edit: I see near surface winds in figure 9, but there is little reference to this figure and the figure needs some improvement for readability.

We have made the figure more readable by replacing curly vectors with straight vectors and removed precipitation fields since those are already shown in Figure 3. In addition, we have added P-E and wind for all scenarios and discussed this figure in more detail.

• A much more careful consideration of the 12xCO2 (and 6xCO2?) simulation is in order, particularly considering the consequences for ECS and overturning. Despite 2000 years of simulation, the final state may still be highly dependent on the initial conditions implying very warm low-latitude intermediate waters.

By adding a new figure that includes the time series of surface and deep-water pot. temperature we have addressed the adjustment of temperature in the ocean and a discussion of the causes of the drift. In the discussion, we added a statement that the state

of ocean circulation at 2000 years depends on the initial conditions that were selected to be the same for all Eocene scenarios in this study.

• The conclusion/discussion section is very limited, with the majority focussing on

DWF and ocean ventilation and thus not representing the main findings well.

Although this is in the main title, I found limited information in the results that

support the suggested collapse of the deep-sea circulation.

We have revised the discussion/conclusion sections accordingly to address the reviewer's concerns.

Specific comments:

• The abstract needs some improvement, as it is rather tedious to read and several

comparison statements lack a clear reference.

We have revised the abstract to make it more readable and consistent to the main body of the paper.

• L15: what does PAL stand for?

Corrected to pre-industrial atmospheric $CO_2$ levels.

I see this is clarified on L72, but I would argue the term 'levels' is a bit ambiguous?

This may suggest you are increasing non-CO2 components as well. If this is

following general syntax used in other work, ignore this comment.

Clarified, see above.

• L16: 'equatorial warming' is a bit ill-defined here. In addition, it is unclear to interpret

the 36.9C value without a reference from proxies or e.g. 1x/3xCO2.

We changed the text accordingly from "equatorial warming" to "tropical warming between 15°S and 15°N".

• L62: Is this adjustment shown somewhere? Even after 2000 years, equilibration may

be highly dependent on e.g. initial conditions.

See above, time series have been included in the revision.

• L85: The initial temperature profile leads to some questions;

o If this is used for all experiments, they likely have very different levels of, and potential biases from equilibration, is this accounted for?

We agree with the reviewer that the adjustment to equilibrium depends on the scenario, so we will only discuss the final drift for the different experiments. Limited computations resources as allocated to the project allowed us only to integrate for 2,000 years which is quite typical for CESM paleo studies.

o The linear temperature profile with depth should lead to some very warm intermediate levels, which are physically unrealistic and may lead to large temperature biases lasting many 1000s years in stable regions.

e.g. temperatures at 1000-2000m depth at low latitude are generally 10-15C in similar 3/4xCO2 Eocene simulations, while they would be ~32-36C initially.

I would not expect such a large deviation to dissipate in these simulations.

o Is there a reason to have a discontinuity at z=5000m and scale with 6000m instead?

We corrected the formula for the initial condition of the potential temperature

"The model has been initialized from a depth(z) −dependent temperature profile (with a vertically declining temperature with T=25°C (5000-z)/5000 for temperature $z \leq 5000$ m and 15 °C for z > 5000

Thus, surface temperature-at surface is 25°C and in the abyss 15°C and vertical sections of T do indicate a warm pool at intermediate depth. By adding TS profiles in the revision readers of the manuscript will better understand water mass distributions.

• L89: In contrast to the very warm initialisation, there is a positive net TOA radiative balance (I assume it is defined positive downward?). While still quite large, I would expect negative values considering the initialisation. Some further analysis of the time evolution of e.g. depth-dependent temperature and radiative balance would be

For the revision, we have added time series of temperature at the surface and at 4000 m and discuss the adjustment for the different climate simulations.

• L103: Why would this effect be uniquely important to West-Antarctica, rather than all high-latitude regions?

We have changed the sentence accordingly, The higher-than-pre-industrial surface air temperature in the 1xCO2 Paleocene Baseline experiments leads to a reduced snow height and snow-free areas in polar regions which in turn reduce the reflectivity and amplify the warming, particularly over ice sheet-free West Antarctica.

• L109: This is confusing; global precipitation is linked to polar temperatures rather than warmer SSTs overall? It is also not straightforward to me how land albedo is linked directly to precipitation.

We have deleted "absence of ice sheets, and changes in the land surface albedo"

• L113: Where do we see precipitation differences over land relative to the PI scenario? The argumentation is a bit weak here; why would a strengthened ITCZ have a different effect over land versus ocean?

In this paragraph we have changed the text accordingly by focusing on global pattern and not on difference land vs ocean:

"The precipitation increases in the 1xCO2 experiment relative to the preindustrial conditions is linked to heavy rainfall over the tropics, due to the intensification of moisture transport along the lower branch of the Hadley Cell into the intertropical convergence zone."

• L120: I am missing an explanation on the drying patterns close to the equator over much of the ocean. These do not align with the downward branches of the Hadley cells, so are seemingly related with a migration and/or contraction of the ITCZ which further intensifies at higher CO2? (edit: I see this is the topic of sections 3.2/3.3, but still miss a clear explanation there).

We have clarified the text accordingly by explaining the downward pattern close the ITCZ: "Lower SST at the equator, particularly in the Pacific Ocean are linked due to Ekman-induced upwelling contribute to band of low precipitation."

• L129: This is mostly a repetition of an earlier paragraph. Also, 200Sv is mentioned here versus >18$^0$ above?

We have deleted the sentence accordingly to avoid repetition.

• L145: I do not see SST in Table 1? Is the value the same by coincidence, or a typo?
SST values have been included in Table 1 in the revised version of the paper,

• L149: Is the ice albedo feedback shown somewhere? As sea ice is said to only be present in winter, this would unlikely have a major radiative effect.

We have de-emphasized the ice albedo effect in the revision and mentioned that this will be only a seasonal climate feedback.

• L168: I assume this is about the global average?

Yes, we have clarified the sentence accordingly.

• L171: The decline in latent heat flux is contradictory to what was said earlier (and further down; L180) when considering rainfall, if this is land-only versus global some clarification is needed. More generally, a discussion on the radiative balance would be helpful (which I believe is shown in the general DeepMIP paper including these simulations?). What about the potential role of albedo and clouds? Are the land temperature changes highly seasonally dependant?
We have revised this sentence and included a more detailed analysis.

• L172: This is again land only?

Yes. We have reorganized sentences to discuss global averages first and then changes over land.

• L173: It is a bit counterintuitive in this section to generally treat the 3xCO2 case as a reference, while this is not shown as such in the figures. While it does make sense following up on the previous section, this is not consistent with the figures and

makes the interpretation more difficult. For example: further strengthening of equatorial rains is seen for 6x/12x CO2, consistent with the 3xCO2 anomaly pattern, but there is a clear double ITCZ at 1xCO2. It is therefore difficult to assess whether the former show only a further strengthening of the anomaly pattern, or the background as well.

We have discussed the double ITCZ in the revision and increase in precipitation linked to an increase in convective precipitation with a rise in $CO_2$ radiative forcing.

• L176: Where/how can I see the enhanced moisture transport mentioned here?

We have improved this sentence by including the assumption that under steady state conditions, <TR> = <E>-  where TR denotes the horizontal moisture transport of water vapor.

• L179: An increased downward branch of the Hadley cell is mentioned (but not shown?) explaining drier conditions in the subtropics. Although reasonable, I think the assessment of precipitation patterns is too limited here;

We have revised the sentence accordingly with a more detailed assessment and literature references.

o Overall tropical rains are increasing towards higher T/CO2, but the rain bands are contracting towards the equator.

More intense equatorial rainbands are linked to an increase in convective precipitation (see above)

o Tropical rainbands cover a significantly wider latitude range in the PB versus PI, but contract again towards higher CO2. Can you explain this?

See above – This appears to be linked to an increase in convective precipitation towards higher $CO_2$ radiative forcing.

o While evaporation is higher due to lower saturation in a warmer atmosphere, precipitation does not decrease in the subtropics for higher CO2.

We have improved this paragraph in the revision, discussing changes in global patterns followed by regional changes.

o Is there a significant role of monsoonal rains in understanding changes in

precipitation?

Monsoonal precipitation affects seasonal rainfall patterns that can also affect annual patterns. However, this paper focuses on mean climate and changes in the ocean circulation and not seasonal variations.

However,

o Extratropical rains increase significantly and expand polewards for higher

CO2, while the meridional temperature gradients reduce. What does this

mean for midlatitude storm tracks?

The atmospheric resolution in CESM1.2 is finite volume ~1.9°x2.5° resolution which very likely underrepresents the strength of the storm tracks. A higher resolution simulation of ~0.25° would be more adequate to resolve storm tracks, and we refer to Shields et al. (2021, Palaeogeography, Palaeoclimatology, Paleoecology). We have included this reference in the revision.

• L181: I assume we are looking at figure 5 here? In that case, jumping from zonal wind

stress on the ocean to trade winds is a bit steep. In its current form, the wind

stresses are also quite tough to assess from the figure.

We have cited Figure 5 in L181 and increased the number of labels for zonal wind stress.

• L182: As a reader, it is tough to assess the role of the mentioned albed-related

feedbacks as they are not shown or referred to.

We have clarified the assessment of feedbacks in the revised version.

• L183: Looking at Figure 8, Ekman upwelling seems to be displaced as much as being

reduced? This is, however, hard to see clearly from the figure.

We have improved Figure 8 by smoothing vertical velocity and using non-linear contour labels and revised the text of L183 to better represent changes shown in Figure 8.

• L187: conclusions are made here that deserve more careful consideration; without a

thorough assessment of deep ocean equilibration it is near impossible to have a

clear conclusion on the overturning circulation state.

This has been addressed by including a temperature time series for the surface and 4000m (see above)

Looking at the age tracers in

figure 13 (which are very relevant, but not mentioned up to this point?), my suspicion

is that much of the deep ocean is completely stagnant and therefore dominated by

the initial conditions.

Although the CESM1.2 may have some "memory" of initial conditions the new time series reveal that the various scenarios adjusted to the forcing boundary condition. While the drift appears to be low for scenarios 3x-12x $CO_2$ the drift for the 1xCO2 is highest due to adjustment to boundary conditions that predict a much cooler climate than given initial conditions. The revised manuscript has discussed this issue in more detail.

As the latter are very warm, this run likely has a TOA imbalance

which appears acceptable, but may not at all mean that the ocean is adjusted to the

applied forcing. Despite relatively weak and shallow, the northern overturning cell at

12xCO2 still indicates sinking mostly at 40-60N and down to 1-1.5km. I do not see

how this represents a subtropical haline mode?

In the revision, we have de-emphasized deepwater formation in the subtropics but highlighted a deepening of the mixed layer-depth and an increase in subtropical density by increase in evaporation from the surface for the PETM 6x and 12x $CO_2$ scenarios.

• L191: Again, I miss a clear assessment of the actual ice-related feedbacks here. I

assume the 3xCO2 case only has ice in the wintertime, so any ice albedo-related

feedback should be minimal.

We have clarified the assessment of feedbacks in the revision of the manuscript.

---

## Author Comment (AC3)

We thank the reviewer for the detailed assessment and will address the concerns as listed below

This is a review of "Collapse of deep-sea circulation during an Eocene Hyperthermal Hothouse – A DeepMIP study with CESM1.2" by Winguth and co-authors.

The manuscript is well written and easy to read.

I wonder though what is the real added value of this manuscript. The model results are well described but are mostly a catalogue of known model results and the discussion to the now relatively large literature on the model results of the 1st phase of DeepMIP is very limited. For example, though precipitations occupy a central place in the text and the figures (Figs. 3, 4, 9, 10 and 12), there is no mention or discussion of the paper by Cramwinckel et al. (2023) that precisely focusses on the hydrological cycle in DeepMIP model results, or of the paper by Williams et al. (2022) on African hydroclimate.

In the revision, we have highlighted the importance of this manuscript while focusing on the deep-sea circulation and provide more analysis considering the literature mentioned above.

More importantly, given the title of the manuscript, the case for a collapse of the deep-sea circulation in the simulations described is sloppy at best. The title is misleading because there is no exploration of the mechanisms that make the deep-sea circulation collapse beyond stating that the MOC is less intense. The absence of diagnostics makes it impossible to judge whether the ocean is in "near-equilibrium" (as stated) and fails to provide evidence for the so-called subtropical haline mode that is said to exist at 12x CO2.

In the revision, we have included a time series of pot. temperature at the surface and 4000 m including a computation of the drift for each scenario and compare this drift with the literature. In addition, we have added pot. density at 3000 m to Figure 13 to better assess the change in the deep circulation.

In my opinion, the manuscript should be significantly revised either to provide clear arguments and diagnostics in favour of the collapse of the deep-sea circulation and/or to clearly demonstrate how the results presented contribute further — than what the different manuscripts by Jiang Zhu and colleagues have demonstrated (using the same model with an updated atmosphere), and more generally than what the community has learned from DeepMIP phase 1 — to the understanding of the Early Eocene warmth and/or the PETM.

The revised manuscript better highlights differences to previous publications and includes a more sophisticated analysis of changes in the deep sea-circulation (see above).

Major issues.

There is no way to evaluate whether the ocean is in "near-equilibrium state". In particular, I seriously doubt that the high CO2 simulations are really close to equilibrium but there are no time series of temperature in the manuscript that could be used to check if this is the case in the intermediate and deep ocean. For instance, the CESM1.2 DeepMIP simulations of Zhu and colleagues at 6x and 9x

CO2 in the supplementary materials of Zhang et al. (2022) show that after 2000 years of integration the global mean ocean is definitely warming, and even more so in the intermediate/deep ocean if we assume that the upper ocean is close to equilibrium. I guess the same is happening in your high CO2 simulations.

In the revision, we have included a temperature time series for both surface and deep-sea at 4000 m and have computed the drift to better assess how far the model has adjusted to the forcing material. In addition, a plot of pot. density at 3000 m give us a better quantification of water mass distributions which is compared to the literature listed above.

That the PETM warming generates a transient collapse of the overturning makes perfect sense but out-of-equilibrium snapshot experiments producing a sluggish circulation are in my opinion only poorly supporting this. For instance, with transient simulations, Alexander et al. (2015) nicely show that the PETM MOC collapses during the first few millennia and slowly reinforces until its intensity exceeds the initial pre-perturbation value (Fig. 5 of their supplementary materials). The simulations of Kirtland-Turner et al. (2024) also show this, although on a much smaller scale because the overturning only weakly slows down (their Fig. 3b).

The temperature time series of surface and deep water gives insights into the strength of the overturning series (see above). We discuss the limitations and reference the literature in the revised paper.

Section 1.2

The implementation of DeepMIP conditions should be better explained. It is described for aerosols but not for the other forcings mentioned l. 65-67. For instance:

- what are the differences in implementation between this manuscript and the simulations of Zhu and colleagues reported in DeepMIP (e.g. Lunt et al. 2021)? Notably, Zhu et al. implemented a specific marginal sea balancing scheme for the Arctic Ocean to conserve salinity in their DeepMIP simulations (Lunt et al. 2021, section 2.2.1). Is this also the case here? It might be useful to provide salinity time series as well.

A marginal sea parameterization for the Arctic has been implemented in this study and the model description has been revised accordingly.

- Herold et al. give a river runoff direction map. I would have thought you would use it directly (it was made for the CCSM/CESM model) but your sentence suggests otherwise.

Runoff was taken from a mapping file provided by co-author Christine Shields as part of the paleoclimate version of CESM1.2. We have clarified this in the revision.

- Eccentricity = 0.06 is not preindustrial. Is this an error?

Orbital parameters are selected for 1950 according to Berger 1989 and we corrected this in the revision of the paper.

- how was tidal dissipation implemented?

Tidal dissipation is set as a diagnostic in this simulation, and we clarify the manuscript accordingly.

- the preindustrial solar constant used is 1361 W.m$^{-2}$ but Winguth et al. (2010) uses a PETM solar constant of 1362 W.m$^{-2}$. Was the reference solar constant updated?

We used the preindustrial solar constant of 1361 W m$^{-2}$ in this simulation, which has been clarified in the text.

- it looks like there is a missing minus sign in the initial temperature profile. And should it be 5000 rather than 6000?

 Formula has been corrected accordingly.

Minor points.

l. 15-16. This sentence is correct but not easy to follow. Please reformulate.

We have revised this sentence accordingly.

l. 26, 189, 293. What do you mean by subtropical haline shallow mode exactly? The deepest MLD in the Northern Hemisphere in the 12x CO2 simulation is found westward of Greenland, as in the other simulations at lower CO2 (Fig. 6).

See above; in the revision we have included density at $\sigma$ at 3000m to provide more insights into the water masses. Although mixed-layer depth deepens in the subtropics due to an increase in sea surface salinity in the 6xCO$_2$ and 12xCO$_2$ scenarios we have de-emphasized the haline mode throughout the revision.

l. 55-56. Perhaps use some of the diagnostics shown in Zhang et al. (2022) seeing as this paper analyzes the Early Eocene ocean circulation in DeepMIP models.

The revised manuscript has been better compared with Zhang et al. (2022).

l. 75. Easiest said that the Eocene ice sheet boundary condition is no ice.

We have corrected this statement accordingly.

l. 94-96. Remove. That the Eocene aerosol forcing has negligible effect has already been said, plus the simulation PB_PR is actually not shown anywhere.

We have removed this sentence.

l. 137. Unclear why the authors states that the simulated MOC is consistent with previous studies. To the least, Goldner et al. (2014) and Toumoulin et al. (2020) do not simulate bipolar deep water formation and Eslworth et al. (2017) may do so (though it is hard to state for sure based on the figures) but only with a deep Drake Passage.

In the revision, we have reassessed the agreements and disagreements with findings from the literature.

Data availability. Standard good practice today has it that the data should be open-access. Please provide the outputs the replicate the results.

We will provide open-access to data as soon as this paper is approved for publication in this journal.

Figure 10b is not used in the text.

In the revision, we have cited Figure 10b.

Figure 13. At which depth is shown the ideal age? Also, the arrows are hard to catch.

Depth is at 3000 m and we have improved the figure and figure caption accordingly. We changed also the color of the arrows.